# Cross-sectional study of association between psychosocial stressors with chronic kidney disease among migrant and non-migrant Ghanaians living in Europe and Ghana: the RODAM study

David Nana Adjei,[1,2] Karien Stronks,[1] Dwomoa Adu,[3] Erik Beune,[1] Karlijn Meeks,[1] Liam Smeeth,[4] Juliet Addo,[4] Ellis Owusu-Dabo,[5] Kerstin Klipstein-Grobusch,[6,7] Frank Mockenhaupt,[8] Matthias Schulze,[9] Ina Danquah,[9,10] Joachim Spranger,[11,12,13] Silver Karaireho Bahendeka,[14] Charles Agyemang[1]

For numbered affiliations see end of article.

**Correspondence to**
Dr David Nana Adjei;
dna@chs.edu.gh

## ABSTRACT

**Objectives** The association between psychosocial stressors (PS) and chronic kidney disease (CKD) among sub-Saharan African (SSA) populations is unknown. We examined the association between PS and CKD prevalence among rural and urban Ghanaians and Ghanaian migrants living in three European cities. We also assessed if the influence of PS on CKD is partially mediated by primary risk factors (hypertension and diabetes) of CKD.

**Design** A multi-centred cross sectional data from the Research on Obesity and Diabetes among African Migrants study.

**Setting** Rural and urban Ghana and three European cities (Amsterdam, Berlin and London).

**Participants** A random sample of 5659 adults (Europe 3167, rural Ghana 1043 and urban Ghana 1449) aged 25–70 years.

**Explanatory measures** PS defined by negative life events, perceived discrimination, perceived stress at work/home and depressive symptoms. Three CKD outcomes were considered using the 2012 Kidney Disease: Improving Global Outcomes severity of CKD classification. Comparisons between PS and CKD outcomes were made using logistic regression analyses across all sites.

**Results** We observed higher proportion of negative life events (68.7%) and perceived permanent stress (15.9%) among Ghanaians living in Ghana than Ghanaians living in Europe. Depressive symptoms (7.5%) and perceived discrimination (29.7%) were more common among Ghanaians living in Europe than Ghanaians living in Ghana. No significant association was observed between any of the PS constructs and CKD outcomes across sites except for positive association between stress at work/home and albuminuria (2.81, 95% CI 1.46 to 5.40) and CKD risk (2.78, 95% CI 1.43 to 5.43) among Ghanaians living in Berlin.

**Conclusion** Our study found a positive association between stress at work/home and albuminuria and CKD risk. There was no convincing evidence of associations between the other PS constructs and the prevalence of CKD risk. Further studies are needed to identify potential

## Strengths and limitations of this study

► This study used all three categories of chronic kidney disease (CKD) definitions (albuminuria, reduced estimated glomerular filtration rate and CKD risk) by Kidney Disease: Improving Global Outcomes 2012 in assessing association of psychosocial stressors (PS) with CKD across all sites. This provided more detailed information on CKD outcomes.

► All sites in our study used well-standardised study protocols and this eliminated intraprotocol variability.

► The use of four constructs of PS (negative life events, perceived discrimination, perceived stress at work/home and depressive symptoms) provided a robust approach to assess PS associations with CKD.

► The limitation of intralaboratory variability in earlier studies was eliminated using the same standard operating procedures in the same laboratory for running all samples for all sites.

► PS is captured and experienced in different magnitude across different populations. We were unable to ascertain if PS as defined in this study was adequately captured among Ghanaians living in rural and urban Ghana.

factors driving the high prevalence of CKD among these populations.

## INTRODUCTION

Worldwide, chronic kidney disease (CKD) is a leading cause of mortality and morbidity with several risk factors (diabetes mellitus, obesity, hypertension and cardiovascular disease).[1] The epidemiological transition in low-income and middle-income countries shows increased burden of these risk factors.[2–4] CKD's high morbidity and mortality is mostly driven by uncontrolled comorbidities such as diabetes

and hypertension.[5 6] CKD treatment and management cost is very high and not sustainable even in high-income countries and this underscores the need for prevention.[7] Available literature has shown that both individual and community level economic factors influence CKD.[8–10] However, after adjusting for both individual and community level socioeconomic position, differences in CKD risk among different populations remained.[8 10 11] These findings seem to suggest that other social environmental factors may be driving CKD prevalence and progression in high-risk populations. Recent studies have shown that whereas Ghanaians living in Europe have a higher CKD risk compared with their host nation populations, they have a lower CKD risk compared with their peers living in urban Ghana.[12] The increased risk of CKD observed in urban Ghana was not fully explained by conventional risk factors[12] and socioeconomic status.[13] This underscores the need to identify other modifiable risk factors of CKD for prevention, optimum treatment and efficient management.

Evidence shows that where an individual works or stays influences his or her physiological well-being leading to an increased risk of chronic diseases.[14 15] Thus, migrants' physiological well-being is influenced by the environment (host nations) they move to stay. The association between psychosocial stressors (PS) and CKD as well as the biological pathways through which PS influences CKD progression is poorly understood and complex,[5] although several pathways have been suggested.[16 17] Specifically, PS have been reported to be associated with alteration in the sympathetic/autonomic nervous system activity leading to higher rates of traditional risk factors of CKD including hypertension and diabetes.[18–20] Environmental stressors have been reported to contribute to the development of insulin resistance, metabolic syndrome, obesity and diabetes, which if uncontrolled leads to CKD incidence.[21 22] Other studies[23 24] have suggested that stress attributable to social and/or economic disadvantage is associated with CKD development and progression through an interaction between other psychosocial factors and comorbid behaviours such as alcohol, tobacco and drug use.[25] In addition, undernutrition due to stress adversely impact on fetal environment by impeding fetal growth leading to low birth weight, which has been shown to be associated with CKD in adult life.[25 26]

However, studies linking PS to CKD prevalence and progression vary greatly among different geographical populations.[5 16 27–31] Specifically, in the USA whereas no association was found between PS and CKD,[31–33] another study reported lower prevalence of CKD was associated with greater life stressors at baseline.[31] In contrast, in the Netherlands depressive and anxiety symptoms were observed to be common among patients with CKD and such patients had increased risk of poor clinical outcomes.[28] Similarly, a study conducted in Korea reported a positive relationship between depressive symptoms and CKD.[27] These observations suggest differential impact of PS at different geographical locations.

For example, discrimination among migrants may differ greatly between host population and from their sub-Saharan African (SSA) compatriots. Specifically, some studies have reported differences in PS among rural and urban populations.[34]

Current literature on the association between PS and CKD among SSA populations and their migrants in Europe is scanty and uncertain. We therefore sought to determine the association between PS and CKD prevalence among Ghanaians in rural and urban Ghana and their migrants living in three European cities. Furthermore, we examined the influence of PS on risk factors (obesity, diabetes and hypertension) of CKD.

## METHODS
### Study population and study design
For this study, data from the Research on Obesity & Diabetes among African Migrants (RODAM) study, a multicentre cross-sectional study, were used. The rationale, conceptual framework, design and methodology of the RODAM study have been described in detail elsewhere.[12 13 35 36] To summarise, the study was conducted from 2012 to 2015. Ghanaians aged 25–70 years living in rural and urban communities in Ghana as well as in three European cities (Amsterdam, Berlin and London) were included in this study. We standardised data collection across all sites. Informed consent was obtained from each participant prior to enrolment in the study. In Ghana, participants were randomly drawn from a list of 30 enumeration areas in the Ashanti region based on the 2010 population census. These enumeration areas came from both rural areas and two purposively selected urban cities (Kumasi and Obuasi). For Ghanaians in Amsterdam, we randomly drew participants from the Municipal register. This register holds data on country of birth of citizens and their parents, thus allowing for sampling based on the Dutch standard indicator for ethnic origin. London lacked a population register for migrant groups. Thus, Ghanaian organisations served as sampling frame for the study. Lists of these organisations were obtained from the Ghanaian Embassy and the Association of Ghanaian Churches in the UK in the boroughs known to have the greatest concentration of Ghanaians. Members were selected from the lists of all members of these organisations. In Berlin, the registration office of the federal state of Berlin provided a list of Ghanaian individuals in Berlin but this resulted in low response rate. Because of this, a change was made to use lists of Ghanaian churches and organisations as the sampling frame. Across all sites in Europe, all selected participants were sent a written invitation combined with written information (information sheet) regarding the study and a response card. The participants were contacted by phone to schedule a date and location of the interview with a trained research assistant or opt for the self-administration of the paper questionnaire or digital online version depending on the preference of the participant. After the completion of the

questionnaire, a date for physical examination was then scheduled after a positive response. The participants were instructed to fast from 22:00 hours the night before the physical examination. The response rate was 76% in rural Ghana and 74% in urban Ghana. In London, of those individuals who were registered in the various Ghanaian organisations and were invited, 75% agreed and participated in the study, while in Berlin, this figure was 68%, and 53% in Amsterdam. For the current study, 5898 participants with data available on both questionnaire data and physical measurements were used. Individuals who were outside the age range of 25–70 years (n=239) were excluded because not all the study sites had individuals outside this age range resulting in 5659 individuals comprising 2492 from rural and urban Ghana and 3167 from the three European cities. In the conduct of analysis, we further excluded individuals with no data on CKD and all other indicators (n=52), resulting in a data set of 5607 participants for analysis.

## MEASUREMENTS
### Covariates
#### Demographic and lifestyle factors
For this study, we obtained information on demographics, educational level and lifestyle factors (smoking and physical activity) by questionnaire. Physical examinations were performed across all sites using validated devices per standardised operational procedures. Educational level was based on the highest qualification gained either in the Netherlands or in the country of origin and was classified into four groups: those who have never been to school or had elementary schooling only, those with lower vocational schooling or lower secondary schooling, those with intermediate vocational schooling or intermediate/higher secondary education schooling and those with higher vocational schooling or university. The four categories were further categorised into three categories by combining the second and third categories. Smoking status was determined from the response to the question "Do you smoke at all?" and was classified into non-smokers and current smokers. Physical activity was assessed using the WHO Global Physical Activity Questionnaire V.2. Weight was measured in light clothing and without shoes with SECA 877 scales to the nearest 0.1 kg. Height was measured without shoes with a portable stadiometer (SECA 217) to the nearest 0.1 cm. Body mass index (BMI) was calculated as weight (kg) divided by height squared ($m^2$). Overweight was defined as BMI of 25 to <30 kg/$m^2$ and obesity as BMI ≥30 kg/$m^2$. Waist circumference was measured in centimetres at the midpoint between the lower rib and the upper margin of the iliac crest. We used the same assessor for each participant in measuring all anthropometrics and each was measured twice; the average of the two measurements was used for analyses.

### Predictor: PS
For this study, four constructs of psychosocial stress (discrimination, perceived stress at work or at home, negative life events and depressive symptoms) were used as explanatory variables.

#### Perceived discrimination
Everyday discrimination as perceived by participants was reported as routinely experiencing instances of unfair treatment. We used the Everyday Discrimination Scale (EDS). The EDS comprises nine items, which rates the frequency at which participants experience daily mistreatment and it focuses on being treated with less courtesy or less respect, receiving poorer service than other people or being called names or insulted. Participants had the option of rating each of the nine items from 'never'=1 to 'very often'=5. The obtained scores were summed and an average of the scores was computed to obtain a final score of 1–5. This scale was used because it is commonly used for self-reported discrimination,[37] with consistent high reliability among a variety of ethnicities,[38] comprising African migrants in the Netherlands.[39]

#### Perceived stress at work or at home
We defined perceived stress at work or at home as 'sense of irritation, filled with anxiety, or as having difficulties in sleeping because of circumstances at work or at home'. We used the psychological stress scale created by the INTERHEART study.[40] Participants in the study were asked about their opinion on frequency of stress at work and at home, and could answer 'never', 'some periods', 'several periods' or 'continually'. Both answers were then combined into a composite score and graded into four categories: never experienced to experienced permanent stress at home or at work.[40] Due to the very small numbers in the permanent periods of stress group, we combined experienced several periods of stress at home or at work and permanent periods of stress at home or at work.

#### Negative life events
The presence of major negative life events among participants was perceived as any event that could cause acute stress to an individual. We therefore applied the well-validated and widely used list of threatening experiences (LTE).[41 42] The scale comprised 12 unpleasant events participants perceived to have experienced in the past 12 months. We used a slightly altered version of LTE consisting of nine unpleasant items. We dichotomised participants into two groups namely 'no negative life events' and 'one or more events' and participants in the second category were expected to have higher levels of stress.[42]

#### Depressive symptoms
Depressive symptoms were measured by the nine-item Patient Health Questionnaire (PHQ-9). The PHQ-9 consists of nine items, with a response scales 0 'not at all', 1 'on several days', 2 'on more than half of the days' and 3 'nearly every day'. A participant was considered to be

in a significant depressed mood when one or both of the items 1 (little interest or pleasure in doing things) and 2 (feeling down, depressed or hopeless) were answered with at least 'on more than half of the days', and at least five of the nine items were answered with at least 'on more than half of the days'.[43]

## Comorbidity factors

Blood pressure (BP) was measured three times using a validated semi-automated device (The Microlife WatchBP home) with appropriate cuffs in a sitting position after at least 5 min rest. The mean of the last two BP measurements was used in the analyses. Hypertension was defined as systolic BP 140 mm Hg and/or diastolic BP 90 mm Hg, and/or being on antihypertensive medication treatment, and/or self-reported hypertension. Trained research assistants in all sites collected fasting venous blood samples according to standard operation procedures, and then temporarily stored at the local research centres. The stored blood samples from the local research centres were transported to Berlin, Germany, according to standardised procedures, for biochemical analyses. This was done to avoid intralaboratory variability. Fasting plasma glucose concentration was measured using an enzymatic method (hexokinase). We defined type 2 diabetes according to the WHO diagnostic criteria (fasting glucose 7.0 mmol/L, and/or current use of medication prescribed to treat diabetes, and/or self-reported diabetes).[44] We assessed concentration of total cholesterol using colorimetric test kits. All biochemical analyses were performed using an ABX Pentra 400 Chemistry Analyzer (ABX Pentra; Horiba 90 ABX, Germany). Hypercholesterolaemia was defined as total cholesterol level ≥6.22 mmol/L. Serum creatinine concentration (in mol/L) was determined by a kinetic colorimetric spectrophotometric isotope dilution mass spectrometry calibration method (Roche Diagnostics).

## Outcome: CKD prevalence

We asked participants to bring an early morning urine sample for the analyses of albuminuria and creatinine levels. Urinary albumin concentration (in µmol/L) was measured by an immunochemical turbidimetric method (Roche Diagnostics). Urinary creatinine concentration (in µmol/L) was measured by a kinetic spectrophotometric method (Roche Diagnostics). Extensive quality checks were done inclusive of blinded serial measurements. Estimated glomerular filtration rate (eGFR) was calculated using the CKD Epidemiology Collaboration creatinine equation.[45] Urinary albumin-creatinine ratio (ACR; expressed in mg/mmol) was calculated by taking the ratio between urinary albumin and urinary creatinine. eGFR and albuminuria were categorised according to the 2012 Kidney Disease: Improving Global Outcomes (KDIGO) classification.[46] eGFR was categorised as follows: G1, 90 mL/min/1.73 m$^2$ (normal kidney function); G2, 60–89 mL/min/1.73 m$^2$ (mildly decreased); G3a, 45–59 mL/min/1.73 m$^2$ (mildly to moderately

decreased); G3b, 30–44 mL/min/1.73 m$^2$ (moderately to severely decreased); G4, 15–29 mL/min/1.73 m$^2$ (severely decreased) and G5, <15 mL/min/1.73 m$^2$ (kidney failure). Albuminuria categories were derived from ACR and were as follows: A1, <3 mg/mmol (normal to mildly increased); A2, 3–30 mg/mmol (moderately increased) and A3, >30 mg/mmol (severely increased). CKD risk was categorised according to severity of kidney disease (green, low risk; yellow, moderately increased risk; orange, high risk and red, very high risk) using the combination of eGFR (G1–G5) and albuminuria (A1–A3) levels defined by the 2012 KDIGO guidelines.[47] Due to the small number of participants in the very high-risk category of CKD (n=27), the high and very high-risk groups were combined. Because of the small number of participants in the severely increased albuminuria category (A3, n=62), we defined albuminuria as ACR 3 mg/mmol by combining the moderately increased (A2) and severely increased (A3) categories.

Covariates assessed were age, sex, educational level and length of stay in Europe. Length of stay was assessed for Ghanaian migrants only. Length of stay was defined as the number of years lived in Europe at the time of data collection. Length of stay was controlled for due to evidence suggesting that it influences mental health.[48] Other covariates were hypertension, obesity and diabetes.

## Patient and public involvement

Community leaders were involved in the recruitment of patients. These comprised religious communities (churches and mosques), endorsement from local key leaders and establishing relationships with healthcare organisations. We also provided information on the study by involving the local media (radio and television stations). We sent letters to all selected health and community authorities to notify participants of the study. Team members were sent to the various communities to stay among the community and organise mini clinics for a period of 1–2 weeks. Results of the study were disseminated through seminars, durbars and via radio and television stations.

## Statistical methods

Characteristics of participants were expressed as absolute numbers and percentages for categorical variables and means and SD for continuous variables. The z-test for proportions was used to compare proportions of demographic and clinical variables among the various sites and the independent t-test was also used to test for mean differences between the two sites. ORs and their corresponding 95% CIs were estimated by means of binary logistic regression analyses to study the associations of albuminuria (ACR >3 mg/mmol, A2–A3, moderately to severely increased albuminuria), reduced kidney function (eGFR <60 mL/min/1.73 m$^2$, G3–G5 moderately to severely decreased kidney function) and increased CKD risk (high and very high CKD risk), with adjustments for covariates.[49] The Spearman's correlation was used to test

for associations between all four constructs of PS. Three models were used to examine the data. Model 1 was adjusted for age and sex, model 2 was adjusted for age and sex and educational level for Ghanaians living in SSA while age, sex, educational level and length of stay for Ghanaians living in Europe.[50–52] Model 3 was adjusted for sex, age, educational level and conventional risk factors (hypertension, diabetes, hypercholesterolaemia, BMI, physical activity and smoking status) of CKD. The analyses were performed for all four constructs of PS using individuals who have not experienced either of the PS per outcome as reference. All tests were stratified per sites due to interactions, Ghanaians living in SSA and Europe; Ghanaians living in rural and urban Ghana and Ghanaians living in Amsterdam, Berlin and London due to an observed interaction between PS and site. Furthermore, the analyses were stratified for those with and without obesity, diabetes, hypertension across all sites due to interactions between these disease risks. P values <0.05 were interpreted as statistically significant. All analyses were performed using STATA, V.14.0 (StataCorp).

## RESULTS
### Characteristics of the study population
Participants characteristics are shown in table 1. Ghanaians living in Ghana were significantly older than their peers living in Europe (47.7±11.9 vs 46.6±9.9, p=0.006). There were more female participants in the Ghana sample compared with European sample (67.1% vs 58.5%, p=0.001). Ghanaians living in Ghana were significantly less educated than those living in Europe. Higher proportion of Ghanaians living in Ghana had experienced negative life events in the last 12 months compared with their peers living in Europe (68.7% vs 59.0%, p=0.001). More than half of Ghanaians living in Ghana had experienced some stress at home or work whereas only a third of those living in Europe had experienced some stress at home or work (p=0.001). Permanent stress at home/work was fairly the same among Ghanaians living in SSA and Europe. Perceived discrimination was significantly higher among Ghanaians living in Europe compared with their peers living in Ghana (29.7% vs 4.8%, p=0.001). Depressive symptoms were more prevalent among Ghanaians living in Europe 7.5% compared with their peers living in Ghana 5.1%. Almost all Ghanaians living in Europe were first-generation migrants. Ghanaians in Europe were significantly more obese, more likely to smoke and less physically active compared with their peers living in Ghana. Prevalence of hypercholesterolaemia was significantly higher, but type 2 diabetes and hypertension were significantly lower among Ghanaians living in Ghana compared with their peers living in Europe (p=0.001). Prevalence of albuminuria, reduced eGRF and CKD risk were higher in Ghanaians living in Ghana compared with those living in Europe.

### Association between PS and CKD
Figures 1-4 show CKD prevalence by negative life events in the past 12 months among Ghanaians living in Ghana and Europe. In Europe, CKD prevalence was fairly the same among those who had experienced any negative life events compared with those who had not in the last 12 months. Prevalence of CKD was higher among Ghanaians who had not experienced any negative life events in the past 12 months (10.9%) compared with those who had experienced some negative life events (9.9%) and living in Ghana. CKD prevalence was higher among Ghanaians who had not experienced any form of discrimination (10.6%) than those who had (6.7%) in Ghana as well as in Europe (figure 2). CKD prevalence was slightly higher among Ghanaians who had experienced several/permanent stress at work/home in the past 12 months and living in Ghana (10.4%) or Europe (9.7%) (figure 3). Ghanaians who did not report any form of depressive symptoms had a significantly higher CKD prevalence than those who did and living in Ghana (10.4%) and Europe (8.7%) (figure 4).

Table 2 shows the correlation matrix between the four constructs of PS among Ghanaians living in Ghana and those living in Europe. All four constructs of PS were positively correlated with each other among Ghanaians living in Europe and Ghanaians living in Ghana (p<0.001), except stress at work/home and discrimination among Ghanaians living in Ghana.

Table 3 shows association between all four constructs of PS and CKD among Ghanaians living in Ghana and those living in Europe. There was no statistically significant association between PS and albuminuria, reduced eGFR and CKD risk among Ghanaians living in Ghana and those living in Europe except individuals living in Europe with some stress and lower risk of reduced eGFR (0.46, 95% CI 0.24 to 0.88). Online supplementary table S1 shows further adjustments for conventional risk factors of CKD. This did not show any statistically significant associations between PS and albuminuria, reduced eGFR and CKD risk among Ghanaians living in Ghana and Europe. Online supplementary table S2 shows further stratification based on obesity status. We did not find any association between PS and CKD for obese participants and those who were not obese for Ghanaians living in rural and urban Ghana and those living in Europe. However, we observed an inverse association between PS and CKD among migrants who were not obese but have experienced discrimination for the past 12 months (0.63 95% CI 0.41 to 0.97). In online supplementary table S3, we stratified our analysis based on diabetic status. We did not find any associations between PS and CKD for participants with and without diabetes for both Ghanaians living in rural and urban Ghana and their migrant peers in Europe. Finally, online supplementary table S4 stratified analysis by hypertension status. No associations were observed between PS and CKD for individuals who had hypertension and those who did not have hypertension for both those living in rural and urban Ghana as well as in

**Table 1**  Baseline characteristics of respondents

| | Ghanaians (SSA) n (%) | Ghanaians (Europe) n (%) | P value |
|---|---|---|---|
| N | 2492 (44.1) | 3167 (55.9) | 0.001 |
| Female sex | 1672 (67.1) | 1851 (58.5) | 0.001 |
| Age (years) | 47.7±11.9 | 46.6±9.9 | 0.006 |
| Educational status | | | |
| Low | 1169 (49.2) | 635 (21.8) | 0.001 |
| Middle | 858 (36.1) | 1111 (38.1) | |
| High | 347 (14.6) | 1168 (40.1) | |
| Negative life events in the past 12 months | | | |
| No | 739 (31.3) | 1158 (41.0) | 0.001 |
| Yes | 1619 (68.7) | 1667 (59.0) | |
| Perceived stress at home/work | | | |
| Never | 692 (29.4) | 1371 (48.8) | 0.001 |
| Some periods | 1290 (54.7) | 1033 (36.8) | |
| Several/Permanent | 375 (15.9) | 407 (14.4) | |
| Perceived discrimination | | | |
| No | 2065 (95.2) | 1960 (70.3) | 0.001 |
| Yes | 104 (4.8) | 829 (29.7) | |
| Depressive symptoms | | | |
| No | 2239 (94.9) | 2582 (92.5) | 0.001 |
| Yes | 119 (5.1) | 209 (7.5) | |
| Migration generation | | | |
| First | Not applicable | 2868 (98.7) | Not applicable |
| Second | Not applicable | 38 (1.3) | Not applicable |
| BMI | | | |
| Normal (<25 kg/m$^2$) | 1373 (55.2) | 643 (20.4) | 0.001 |
| Overweight (25≤30 kg/m$^2$) | 684 (27.5) | 1350 (42.8) | |
| Obese (>30 kg/m$^2$) | 432 (17.3) | 1163 (36.8) | |
| Currently smoking | 36 (1.5) | 121 (4.1) | 0.001 |
| Physical activity | 1255 (52.8) | 1131 (44.0) | 0.001 |
| Hypacholesterolemia | 352 (14.2) | 354 (11.3) | 0.007 |
| Type 2 diabetes | 206 (8.3) | 444 (14.0) | 0.001 |
| Hypertension | 837 (33.6) | 1801 (56.9) | 0.001 |
| ACR | | | |
| A1<3 mg/mmol | 2215 (90.2) | 2814 (91.8) | 0.001 |
| A2–A3≥3 mg/mmol | 243 (9.8) | 252 (8.2) | |
| eGFR | | | |
| G1–G2≥60 mL/min/1.73 m$^2$ | 2377 (96.3) | 2936 (97.4) | 0.018 |
| G3a–G5<60 mL/min/1.73 m$^2$ | 85 (3.7) | 78 (2.6) | |
| CKD risk | | | |
| Low risk | 2197 (89.6) | 2705 (91.5) | 0.015 |
| Moderate-to-very high risk | 256 (10.4) | 252 (8.5) | |

A1, normal to mildly increased; A2–A3, moderately increased to severely increased; ACR, albumin creatinine ratio; BMI, body mass index; CKD, chronic kidney disease; eGFR, estimated glomerular filtration rate; G1–G2, normal to high kidney function to mildly decreased; G3a–G5, mildly to moderately decreased to kidney failure; N, number of respondents; SSA, sub-Saharan Africa.

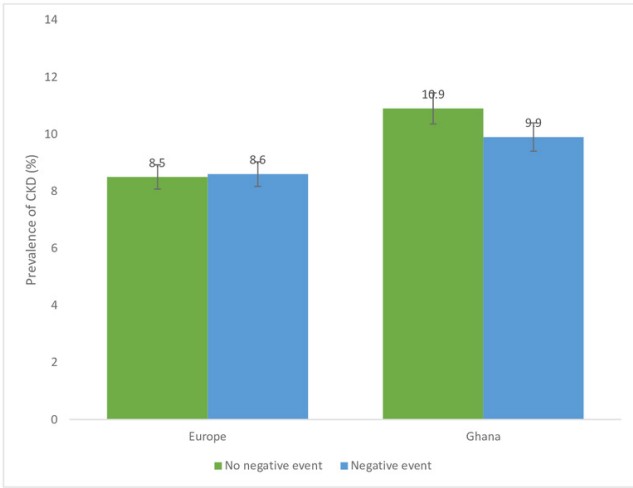

**Figure 1** Prevalence of chronic kidney disease (CKD) risk among Ghanaians who have experienced negative life events and those who have not experienced any negative life events stratified by site. Definition per 2012 Kidney Disease: Improving Global Outcomes guidelines.

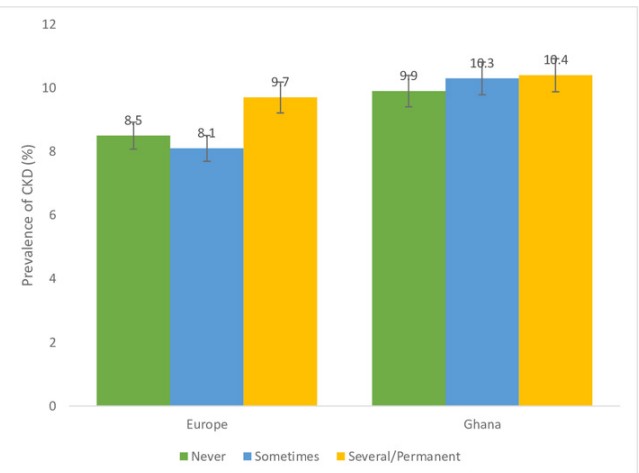

**Figure 3** Prevalence of chronic kidney disease (CKD) risk among Ghanaians who have experienced stress at home/work and those who have not experienced any stress at home/work in the past 12 months stratified by site. Definitions per 2012 Kidney Disease: Improving Global Outcomes guidelines.

their compatriots living in Europe. An inverse association was observed between PS and CKD among Ghanaians who have experienced discrimination in the last 12 months with no hypertension and living in Europe (0.51, 95% CI 0.27 to 0.97). Also, we observed that having experienced some stress at home/work was inversely associated with reduced eGFR among Ghanaians with hypertension and living in Europe (0.47, 95% CI 0. 0.23 to 0.95).

Table 4 shows associations between all 4 constructs of PS and CKD stratified by Ghanaians living in urban and rural Ghana. There was no association between PS and albuminuria, reduced eGFR and CKD risk among Ghanaians living rural and urban Ghana.

Table 5 shows associations between all four constructs of PS and CKD stratified by Ghanaians living in Amsterdam, Berlin and London. There were no associations between PS and albuminuria, reduced eGFR and CKD risk among Ghanaians living in Europe except for positive association between stress at work/home and albuminuria (2.81, 95% CI 1.46 to 5.40) and CKD risk (2.78, 95% CI 1.43 to 5.43) among Ghanaians living in Berlin.

## DISCUSSION
### Key findings
Whereas there was an association between those who have experienced some stress at home/work and reduced

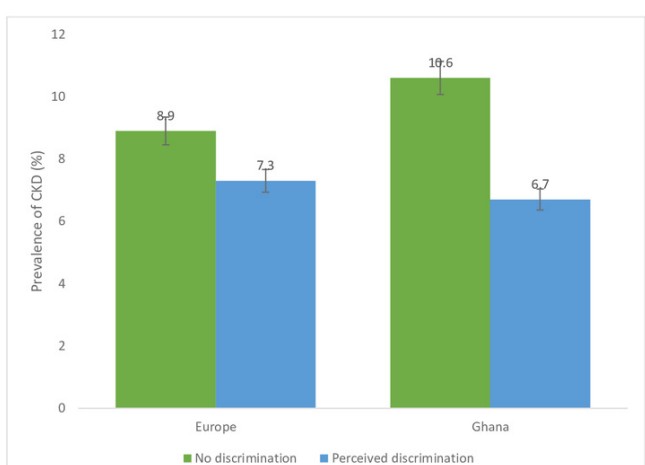

**Figure 2** Prevalence of chronic kidney disease risk among Ghanaians who have experienced discrimination and those who have not experienced any discrimination stratified by site. Definitions per 2012 Kidney Disease: Improving Global Outcomes guidelines.

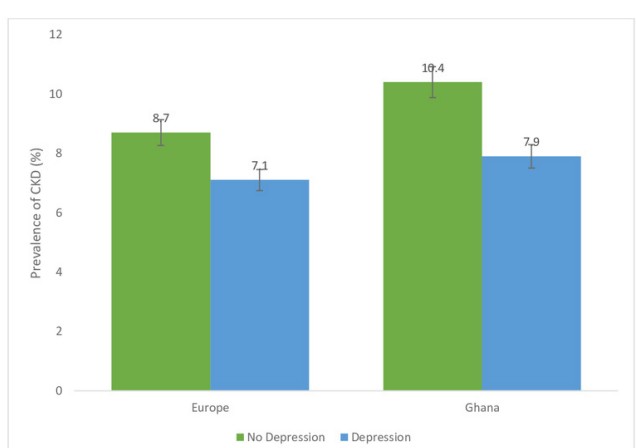

**Figure 4** Prevalence of chronic kidney disease (CKD) risk among Ghanaians who have depressive symptoms and those who do not have depressive symptoms stratified by site. Definitions per 2012 Kidney Disease: Improving Global Outcomes guidelines.

**Table 2** Relationship between PS constructs (negative life events, discrimination, stress at work or home and depression) by Ghanaians (SSA) and Ghanaians (Europe)

| Correlation matrix | Negative events | Discrimination | Stress at work/home | Depression |
|---|---|---|---|---|
| Europe | | | | |
| Negative events | 1.000 | | | |
| Discrimination | 0.152* | 1.000 | | |
| Stress at work/home | 0.297* | 0.161* | 1.000 | |
| Depressive symptoms | 0.143* | 0.136* | 0.285* | 1.000 |
| Ghana | | | | |
| Negative events | 1.000 | | | |
| Discrimination | 0.079* | 1.000 | | |
| Stress at work/home | 0.101* | −0.032 | 1.000 | |
| Depressive symptoms | 0.091* | 0.042 | 0.185* | 1.000 |

*Significant at 1%, Spearman's correlation.
PS, psychosocial stressors; SSA, sub-Saharan Africa.

eGFR among Ghanaians living in Europe, we did not find any association between PS and CKD among Ghanaians living in rural and urban Ghana and their peers in Europe. Also, PS was not associated with CKD for those living in rural and urban Ghana and neither for those living in the three European cities. However, there was an association between stress at work/home and albuminuria and CKD risk among Ghanaians living in Berlin. Further adjustment for conventional risk factors of CKD yielded similar results.

### Discussion of key findings
#### Association between PS and CKD in Ghana
Our study did not find any association between any of the four constructs of PS and prevalence of CKD (albuminuria, reduced eGFR and CKD risk) among Ghanaians living in rural and urban Ghana. Our findings are however in contrast with earlier studies which reported positive associations between PS and prevalent of CKD.[28 31 53] Other studies have hypothesised that the influence of PS on CKD may be important in only those with hypertension and diabetes and that PS may mediate or moderate the association between renal functioning and lifestyle behaviours such as smoking and physical activity.[33] For example, they argue that stress enhances sympathetic nervous system (SNS) activity to increase glucocorticoid secretion and inflammatory cytokines, which heavily contribute to hypertension, diabetes and vascular disease, which are major risk factors of CKD incidence and prevalence.[54] The lack of association between PS and CKD in this present study is unclear due to lack of literature on the association between PS and CKD prevalence, particularly in rural and urban populations. Worth noting, however, is the presence of rich family support systems in the Ghanaian context, especially in rural Ghana, which may help individuals with CKD to cope with PS thereby minimising its effect. For example, patients with limited social networks and low social support have been shown

to have augmented risk of morbidity and mortality.[55–57] Specifically, there is evidence that positive social support is a protective factor for persons dealing with long-term disease conditions.[58] Other studies have reported a protective relationship between social networks, emotionally supportive relationship and threats to physiological and psychosocial health.[59]

#### Association between PS and CKD Amsterdam, Berlin and London
Literature on the association between PS and CKD prevalence among migrants is scant and absent in most European populations. The lack of positive association between PS and CKD in our study is consistent with recent studies conducted among African-Americans[31 33] and other populations.[32 60] Specifically, a recent study using data from the Jackson Heart Study, which comprised extensive constructs of psychosocial variables reported that greater life stressors were associated with lower prevalence of CKD at baseline.[31] Several studies in other parts of the world have reported a positive relationship between higher prevalence of stressors and CKD risk,[27 28] although the study findings have been inconsistent. Whereas some did not find any associations among African-Americans,[31] others found associations in other populations. Even among those who found some associations the directions differed.[28] Reasons for the lack of association observed in our study among migrants are not fully understood but may reflect the real-world situation. First, migrants from Ghana practice both nuclear and extended family support system as their peers living in rural and urban, this practice may mitigate the impact of stressors such as unemployment, death of a love one, discrimination, etc. They also belong to several religious organisations such as churches, which provide similar support systems against stressors. Moreover, there are several associations of the various ethnic groups (Akan, Ga and Ewe) providing such support when the need arises. These systems provide both instrumental and/or emotional social support.[61] These

**Table 3** Association of psychosocial stressors (PS) indicators (negative life events, discrimination, stress at work/home and depression) with albuminuria, reduced eGFR and CKD risk for Ghanaians living in Ghana and those living in Europe

| | Albuminuria (ACR≥3mg/mmol) | | | eGFR<60mL/min/1.73m² | | | High-to-very high CKD risk (KDIGO, 2012) | | |
| | | OR (95%CI) | | | OR (95%CI) | | | OR (95%CI) | |
| | n (%) | Model 1 | Model 2 | n (%) | Model 1 | Model 2 | n (%) | Model 1 | Model 2 |
|---|---|---|---|---|---|---|---|---|---|
| **Negative events** | | | | | | | | | |
| Europe | | | | | | | | | |
| No | 1128 (8.2) | 1.00 (Reference) | 1.00 (Reference) | 1106 (2.6) | 1.00 (Reference) | 1.00 (Reference) | 1090 (8.5) | 1.00 (Reference) | 1.00 (Reference) |
| Yes | 1615 (8.4) | 1.03 (0.78 to 1.35) | 1.07 (0.80 to 1.42) | 1587 (2.5) | 0.86 (0.53 to 1.42) | 0.83 (0.49 to 1.39) | 1557 (8.6) | 0.97 (0.76 to 1.32) | 0.99 (0.74 to 1.32) |
| Ghana | | | | | | | | | |
| No | 732 (8.7) | 1.00 (Reference) | 1.00 (Reference) | 736 (4.5) | 1.00 (Reference) | 1.00 (Reference) | 732 (10.9) | 1.00 (Reference) | 1.00 (Reference) |
| Yes | 1595 (3.8) | 0.87 (0.65 to 1.16) | 0.85 (0.63 to 1.14) | 1601 (3.4) | 0.69 (0.45 to 1.08) | 0.67 (0.44 to 1.09) | 1590 (9.9) | 0.88 (0.66 to 1.17) | 0.86 (0.64 to 1.15) |
| **Discrimination** | | | | | | | | | |
| Europe | | | | | | | | | |
| No | 1899 (8.5) | 1.00 (Reference) | 1.00 (Reference) | 1867 (2.6) | 1.00 (Reference) | 1.00 (Reference) | 1832 (8.9) | 1.00 (Reference) | 1.00 (Reference) |
| Yes | 810 (7.4) | 0.87 (0.64 to 1.19) | 0.92 (0.67 to 1.26) | 791 (2.2) | 0.83 (0.47 to 1.47) | 0.84 (0.46 to 1.52) | 782 (7.3) | 0.82 (0.59 to 1.12) | 0.84 (0.60 to 1.16) |
| Ghana | | | | | | | | | |
| No | 2034 (10.0) | 1.00 (Reference) | 1.00 (Reference) | 2047 (3.9) | 1.00 (Reference) | 1.00 (Reference) | 2031 (10.6) | 1.00 (Reference) | 1.00 (Reference) |
| Yes | 104 (7.7) | 0.83 (0.39 to 1.73) | 0.91 (0.67 to 1.24) | 104 (1.9) | 0.67 (0.15 to 2.85) | 0.67 (0.16 to 2.84) | 104 (6.7) | 0.70 (0.32 to 1.55) | 0.71 (0.32 to 1.55) |
| **Stress at home/work** | | | | | | | | | |
| Europe | | | | | | | | | |
| Never | 1330 (8.2) | 1.00 (Reference) | 1.00 (Reference) | 1305 (3.4) | 1.00 (Reference) | 1.00 (Reference) | 1282 (8.5) | 1.00 (Reference) | 1.00 (Reference) |
| Some stress | 1002 (7.9) | 0.97 (0.72 to 1.31) | 1.04 (0.76 to 1.42) | 984 (1.4) | **0.47 (0.26 to 0.87)** | **0.46 (0.24 to 0.88)** | 968 (8.6) | 0.96 (0.71 to 1.30) | 1.02 (0.74 to 1.39) |
| Several/Permanent stresses | 397 (9.1) | 1.11 (0.74 to 1.64) | 1.153 (0.77 to 1.72) | 390 (2.3) | 0.73 (0.35 to 1.52) | 0.76 (0.36 to 1.61) | 383 (9.7) | 1.13 (0.77 to 1.68) | 1.19 (0.80 to 1.79) |
| Ghana | | | | | | | | | |
| Never | 682 (10.3) | 1.00 (Reference) | 1.00 (Reference) | 688 (3.3) | 1.00 (Reference) | 1.00 (Reference) | 682 (9.9) | 1.00 (Reference) | 1.00 (Reference) |
| Some stress | 1279 (9.5) | 0.87 (0.64 to 1.19) | 0.80 (0.59 to 1.11) | 1279 (3.9) | 1.06 (0.63 to 1.77) | 1.11 (0.66 to 1.87) | 1274 (10.3) | 0.95 (0.69 to 1.30) | 0.92 (0.67 to 1.26) |
| Several/Permanent stresses | 365 (8.5) | 0.75 (0.48 to 1.18) | 0.68 (0.59 to 1.11) | 369 (4.1) | 1.13 (0.57 to 2.23) | 1.22 (0.61 to 2.46) | 365 (10.4) | 0.96 (0.63 to 1.47) | 0.92 (0.59 to 1.42) |
| **Depressive symptoms** | | | | | | | | | |
| Europe | | | | | | | | | |
| No | 2505 (8.5) | 1.00 (Reference) | 1.00 (Reference) | 2457 (2.7) | 1.00 (Reference) | 1.00 (Reference) | 2416 (8.7) | 1.00 (Reference) | 1.00 (Reference) |
| Yes | 206 (6.3) | 0.71 (0.39 to 1.27) | 0.76 (0.43 to 1.36) | 202 (1.5) | 0.63 (0.19 to 2.03) | 0.68 (0.21 to 2.23) | 199 (7.1) | 0.78 (0.44 to 1.37) | 0.83 (0.47 to 1.46) |
| Ghana | | | | | | | | | |
| No | 2212 (9.9) | 1.00 (Reference) | 1.00 (Reference) | 2222 (3.8) | 1.00 (Reference) | 1.00 (Reference) | 2207 (10.4) | 1.00 (Reference) | 1.00 (Reference) |
| Yes | 114 (5.3) | 0.45 (0.19 to 1.03) | 0.45 (0.19 to 1.01) | 114 (2.6) | 0.52 (0.16 to 1.72) | 0.53 (0.17 to 1.74) | 114 (7.9) | 0.62 (0.30 to 1.25) | 0.61 (0.30 to 1.24) |

Model 1, adjusted for age and sex; model 2, adjusted for age, sex and educational level for Ghanaians (SSA) and length of stay for those in Europe.
The bold values are significant.
ACR, albumin creatinine ration; eGFR, estimated glomerular filtration rate; CKD, chronic kidney disease; KDIGO, Kidney Disease: Improving Global Outcomes; n, total number of Ghanaians living in Ghana and Europe among the various levels of PS constructs; SAA, sub-Saharan Africa; %, proportion of individuals with CKD among the various levels of PS constructs in Ghana and Europe

**Table 4** Association of PS indicators (negative life events, discrimination, stress at work/home and depressive symptoms) with albuminuria, reduced eGFR and CKD risk among rural and urban Ghana

| | Albuminuria (ACR≥3mg/mmol) | | | eGFR<60mL/min/1.73m² | | | High-to-very high CKD risk (KDIGO, 2012) | | |
|---|---|---|---|---|---|---|---|---|---|
| | | OR (95% CI) | | | OR (95% CI) | | | OR (95% CI) | |
| | n (%) | Model 1 | Model 2 | n (%) | Model 1 | Model 2 | n (%) | Model 1 | Model 2 |
| **Negative events** | | | | | | | | | |
| Urban Ghana | | | | | | | | | |
| No | 477 (11.9) | 1.00 (Reference) | 1.00 (Reference) | 477 (4.4) | 1.00 (Reference) | 1.00 (Reference) | 477 (12.2) | 1.00 (Reference) | 1.00 (Reference) |
| Yes | 912 (10.5) | 0.87 (0.61 to 1.23) | 0.87 (0.61 to 1.24) | 911 (3.4) | 0.73 (0.41 to 1.31) | 0.72 (0.40 to 1.29) | 910 (10.8) | 0.87 (0.61 to 1.24) | 0.87 (0.61 to 1.25) |
| Rural Ghana | | | | | | | | | |
| No | 255 (7.5) | 1.00 (Reference) | 1.00 (Reference) | 259 (4.6) | 1.00 (Reference) | 1.00 (Reference) | 255 (8.6) | 1.00 (Reference) | 1.00 (Reference) |
| Yes | 683 (7.6) | 0.97 (0.56 to 1.69) | 0.94 (0.54 to 1.64) | 690 (3.5) | 0.63 (0.31 to 1.31) | 0.66 (0.32 to 1.37) | 680 (8.8) | 0.93 (0.55 to 1.58) | 0.92 (0.54 to 1.56) |
| **Discrimination** | | | | | | | | | |
| Urban Ghana | | | | | | | | | |
| No | 1326 (11.1) | 1.00 (Reference) | 1.00 (Reference) | 1326 (3.9) | 1.00 (Reference) | 1.00 (Reference) | 1325 (11.4) | 1.00 (Reference) | 1.00 (Reference) |
| Yes | 71 (8.5) | 0.85 (0.36 to 2.00) | 0.89 (0.37 to 2.11) | 71 (2.8) | 1.17 (0.27 to 2.09) | 1.16 (0.27 to 2.06) | 71 (7.1) | 0.69 (0.27 to 1.77) | 0.72 (0.28 to 1.83) |
| Rural Ghana | | | | | | | | | |
| No | 708 (8.1) | 1.00 (Reference) | 1.00 (Reference) | 721 (3.9) | 1.00 (Reference) | 1.00 (Reference) | 706 (9.2) | 1.00 (Reference) | 1.00 (Reference) |
| Yes | 33 (6.1) | 0.79 (0.18 to 3.47) | 0.84 (0.19 to 2.65) | 33 (0.0) | * | * | 33 (6.1) | 0.75 (0.17 to 2.89) | 0.83 (0.19 to 2.65) |
| **Stress at home/work** | | | | | | | | | |
| Urban Ghana | | | | | | | | | |
| Never | 460 (10.9) | 1.00 (Reference) | 1.00 (Reference) | 460 (3.3) | 1.00 (Reference) | 1.00 (Reference) | 460 (10.2) | 1.00 (Reference) | 1.00 (Reference) |
| Some stress | 732 (11.5) | 1.04 (0.71 to 1.51) | 0.91 (0.62 to 1.37) | 730 (4.1) | 1.27 (0.66 to 2.43) | 1.30 (0.67 to 2.51) | 730 (11.8) | 1.13 (0.77 to 1.65) | 1.04 (0.71 to 1.53) |
| Several/Permanent stresses | 197 (9.6) | 0.87 (0.50 to 1.52) | 0.74 (0.42 to 1.02) | 198 (3.5) | 1.17 (0.46 to 2.84) | 1.20 (0.47 to 3.09) | 197 (11.7) | 1.15 (0.68 to 1.71) | 1.05 (0.61 to 1.81) |
| Rural Ghana | | | | | | | | | |
| Never | 222 (9.0) | 1.00 (Reference) | 1.00 (Reference) | 228 (3.5) | 1.00 (Reference) | 1.00 (Reference) | 222 (9.5) | 1.00 (Reference) | 1.00 (Reference) |
| Some stress | 547 (6.9) | 0.69 (0.39 to 1.23) | 0.68 (0.38 to 1.22) | 549 (3.6) | 0.88 (0.38 to 2.07) | 0.92 (0.39 to 2.18) | 544 (8.3) | 0.74 (0.42 to 1.30) | 0.75 (0.42 to 1.31) |
| Several/Permanent stresses | 168 (7.1) | 0.63 (0.30 to 1.37) | 0.60 (0.28 to 1.29) | 171 (4.7) | 1.07 (0.38 to 3.03) | 1.21 (0.43 to 3.46) | 168 (8.9) | 0.71 (0.34 to 1.50) | 0.73 (0.35 to 1.50) |
| **Depressive symptoms** | | | | | | | | | |
| Urban Ghana | | | | | | | | | |
| No | 1336 (11.3) | 1.00 (Reference) | 1.00 (Reference) | 1335 (3.8) | 1.00 (Reference) | 1.00 (Reference) | 1334 (11.5) | 1.00 (Reference) | 1.00 (Reference) |
| Yes | 52 (3.9) | 0.30 (0.07 to 1.25) | 0.30 (0.07 to 1.27) | 52 (1.9) | 0.46 (0.06 to 2.50) | 0.45 (0.06 to 2.13) | 52 (5.8) | 0.44 (0.14 to 1.45) | 0.45 (0.14 to 1.48) |
| Rural Ghana | | | | | | | | | |
| No | 876 (7.7) | 1.00 (Reference) | 1.00 (Reference) | 887 (3.8) | 1.00 (Reference) | 1.00 (Reference) | 873 (8.7) | 1.00 (Reference) | 1.00 (Reference) |
| Yes | 62 (6.5) | 0.67 (0.23 to 1.94) | 0.67 (0.23 to 1.94) | 62 (3.2) | 0.58 (0.13 to 2.56) | 0.61 (0.14 to 2.68) | 62 (9.7) | 0.82 (0.33 to 2.01) | 0.85 (0.34 to 2.09) |

Model 1, adjusted for age and sex; model 2, adjusted for age, sex and educational level.
*No case of CKD and therefore ORs were not calculated.
ACR, albumin creatinine ration; eGFR, estimated glomerular filtration rate; CKD, chronic kidney disease; K DIGO , Kidney Disease: Improving Global Outcomes; n, total number of Ghanaians living in urban and rural Ghana among the various levels of PS constructs; %, proportion of individuals with CKD among the various levels of PS constructs in rural and urban Ghana.

**Table 5** Association of PS indicators (negative life events, discrimination, stress at work/home and depressive symptoms) with albuminuria, reduced eGFR and CKD risk among Ghanaians in three European cities

| | Albuminuria (ACR≥3mg/mmol) | | | eGFR<60mL/min/1.73m² | | | High-to-very high CKD risk (KDIGO, 2012) | | |
|---|---|---|---|---|---|---|---|---|---|
| | | OR (95% CI) | | | OR (95% CI) | | | OR (95% CI) | |
| | n (%) | Model 1 | Model 2 | n (%) | Model 1 | Model 2 | n (%) | Model 1 | Model 2 |
| **Negative events** | | | | | | | | | |
| Amsterdam | | | | | | | | | |
| No | 548 (7.3) | 1.00 (Reference) | 1.00 (Reference) | 534 (2.4) | 1.00 (Reference) | 1.00 (Reference) | 521 (7.5) | 1.00 (Reference) | 1.00 (Reference) |
| Yes | 784 (7.8) | 1.08 (0.71 to 1.63) | 1.18 (0.77 to 1.81) | 764 (2.9) | 1.11 (0.55 to 2.23) | 1.15 (0.55 to 2.37) | 742 (8.0) | 1.06 (0.69 to 1.62) | 1.08 (0.71 to 1.66) |
| Berlin | | | | | | | | | |
| No | 213 (9.9) | 1.00 (Reference) | 1.00 (Reference) | 213 (2.4) | 1.00 (Reference) | 1.00 (Reference) | 213 (10.8) | 1.00 (Reference) | 1.00 (Reference) |
| Yes | 329 (10.9) | 1.12 (0.63 to 1.99) | 1.19 (0.67 to 2.15) | 330 (1.8) | 0.64 (0.19 to 2.17) | 0.61 (0.18 to 2.11) | 329 (9.4) | 0.86 (0.48 to 1.52) | 0.91 (0.51 to 1.63) |
| London | | | | | | | | | |
| No | 367 (8.7) | 1.00 (Reference) | 1.00 (Reference) | 359 (3.1) | 1.00 (Reference) | 1.00 (Reference) | 356 (8.7) | 1.00 (Reference) | 1.00 (Reference) |
| Yes | 502 (7.8) | 0.89 (0.55 to 1.46) | 0.83 (0.49 to 1.41) | 493 (2.2) | 0.68 (0.28 to 1.65) | 0.58 (0.22 to 1.51) | 486 (9.1) | 1.04 (0.64 to 1.68) | 0.99 (0.58 to 1.68) |
| **Discrimination** | | | | | | | | | |
| Amsterdam | | | | | | | | | |
| No | 956 (8.3) | 1.00 (Reference) | 1.00 (Reference) | 935 (2.9) | 1.00 (Reference) | 1.00 (Reference) | 909 (8.5) | 1.00 (Reference) | 1.00 (Reference) |
| Yes | 363 (5.0) | 0.59 (0.34 to 1.00) | 0.59 (0.35 to 1.02) | 349 (2.1) | 0.69 (0.30 to 1.62) | 0.81 (0.34 to 1.91) | 342 (5.9) | 0.69 (0.41 to 1.14) | 0.69 (0.41 to 1.16) |
| Berlin | | | | | | | | | |
| No | 329 (10.0) | 1.00 (Reference) | 1.00 (Reference) | 329 (2.1) | 1.00 (Reference) | 1.00 (Reference) | 329 (10.3) | 1.00 (Reference) | 1.00 (Reference) |
| Yes | 209 (11.0) | 1.11 (0.63 to 1.95) | 1.16 (0.65 to 2.05) | 210 (1.9) | 0.83 (0.24 to 2.93) | 0.82 (0.23 to 2.91) | 209 (9.1) | 0.86 (0.48 to 1.56) | 0.89 (0.49 to 1.63) |
| London | | | | | | | | | |
| No | 614 (7.9) | 1.00 (Reference) | 1.00 (Reference) | 603 (2.5) | 1.00 (Reference) | 1.00 (Reference) | 594 (8.9) | 1.00 (Reference) | 1.00 (Reference) |
| Yes | 238 (7.9) | 1.03 (0.59 to 1.81) | 1.18 (0.65 to 2.15) | 232 (2.6) | 1.29 (0.46 to 3.59) | 1.09 (0.35 to 3.43) | 231 (7.8) | 0.93 (0.53 to 1.63) | 0.98 (0.52 to 1.82) |
| **Stress at home/work** | | | | | | | | | |
| Amsterdam | | | | | | | | | |
| Never | 634 (8.4) | 1.00 (Reference) | 1.00 (Reference) | 622 (3.2) | 1.00 (Reference) | 1.00 (Reference) | 603 (8.0) | 1.00 (Reference) | 1.00 (Reference) |
| Some stress | 478 (5.7) | 0.68 (0.42 to 1.11) | 0.69 (0.42 to 1.13) | 462 (1.9) | 0.64 (0.29 to 1.43) | 0.68 (0.30 to 1.52) | 452 (6.0) | 0.74 (0.45 to 1.20) | 0.74 (0.45 to 1.22) |
| Several/Permanent stresses | 210 (9.1) | 1.09 (0.63 to 1.91) | 1.12 (0.64 to 1.95) | 204 (2.5) | 0.71 (0.26 to 1.95) | 0.77 (0.28 to 2.14) | 198 (10.1) | 1.24 (0.71 to 2.14) | 1.26 (0.73 to 2.20) |
| Berlin | | | | | | | | | |
| Never | 250 (9.0) | 1.00 (Reference) | 1.00 (Reference) | 250 (2.0) | 1.00 (Reference) | 1.00 (Reference) | 250 (6.4) | 1.00 (Reference) | 1.00 (Reference) |
| Some stress | 196 (15.3) | 2.50 (1.33 to 4.71) | 2.81 (1.46 to 5.40) | 197 (1.5) | 0.88 (0.20 to 3.79) | 0.83 (0.19 to 3.62) | 196 (14.8) | 2.57 (1.34 to 4.90) | 2.78 (1.43 to 5.43) |
| Several/Permanent stresses | 96 (10.4) | 1.64 (0.72 to 3.73) | 1.69 (0.73 to 3.91) | 197 (3.1) | 2.10 (0.47 to 9.46) | 2.04 (0.44 to 9.26) | 76 (9.4) | 1.52 (0.65 to 3.58) | 1.58 (0.66 to 3.75) |
| London | | | | | | | | | |
| Never | 446 (9.2) | 1.00 (Reference) | 1.00 (Reference) | 433 (4.4) | 1.00 (Reference) | 1.00 (Reference) | 429 (10.5) | 1.00 (Reference) | 1.00 (Reference) |
| Some stress | 328 (7.0) | 0.73 (0.43 to 1.25) | 0.79 (0.44 to 1.40) | 325 (0.6) | 0.17 (0.04 to 0.73) | 0.09 (0.01 to 0.67) | 320 (6.9) | 0.65 (0.38 to 1.10) | 0.66 (0.37 to 1.19) |
| Several/permanent stresses | 91 (7.7) | 0.81 (0.35 to 1.87) | 0.86 (0.35 to 2.14) | 90 (1.1) | 0.27 (0.03 to 2.12) | 0.24 (0.03 to 2.05) | 74 (8.9) | 0.83 (0.38 to 1.83) | 0.92 (0.39 to 2.16) |

Continued

**Table 5** Continued

| | Albuminuria (ACR≥3mg/mmol) | | | eGFR<60 mL/min/1.73 m² | | | High-to-very high CKD risk (KDIGO, 2012) | | |
| | | OR (95% CI) | | | OR (95% CI) | | | OR (95% CI) | |
| | n (%) | Model 1 | Model 2 | n (%) | Model 1 | Model 2 | n (%) | Model 1 | Model 2 |
|---|---|---|---|---|---|---|---|---|---|
| **Depressive symptoms** | | | | | | | | | |
| **Amsterdam** | | | | | | | | | |
| No | 1199 (7.8) | 1.00 (Reference) | 1.00 (Reference) | 1135 (2.8) | 1.00 (Reference) | 1.00 (Reference) | 1135 (7.9) | 1.00 (Reference) | 1.00 (Reference) |
| Yes | 121 (6.6) | 0.81 (0.39 to 1.72) | 0.83 (0.39 to 1.76) | 118 (1.7) | 0.65 (0.15 to 2.77) | 0.71 (0.16 to 3.06) | 116 (6.9) | 0.83 (0.39 to 1.76) | 0.82 (0.38 to 1.74) |
| **Berlin** | | | | | | | | | |
| No | 503 (10.7) | 1.00 (Reference) | 1.00 (Reference) | 504 (2.2) | 1.00 (Reference) | 1.00 (Reference) | 503 (10.1) | 1.00 (Reference) | 1.00 (Reference) |
| Yes | 34 (5.9) | 0.53 (0.12 to 2.27) | 0.49 (90.11 to 2.13) | 34 (0.0) | * | * | 34 (5.9) | 0.55 (0.13 to 2.37) | 0.52 (0.12 to 2.24) |
| **London** | | | | | | | | | |
| No | 803 (8.3) | 1.00 (Reference) | 1.00 (Reference) | 785 (2.6) | 1.00 (Reference) | 1.00 (Reference) | 778 (8.9) | 1.00 (Reference) | 1.00 (Reference) |
| Yes | 51 (5.9) | 0.67 (0.20 to 2.21) | 0.91 (0.27 to 3.07) | 50 (2.0) | 0.91 (0.11 to 7.43) | 1.15 (0.14 to 9.54) | 49 (8.2) | 0.94 (0.33 to 2.69) | 1.30 (0.44 to 3.81) |

Model 1, adjusted for age and sex; model 2, adjusted for age, sex, educational level and length of stay.
The bold values are significant.
*No case of CKD and therefore ORs were not calculated.
ACR, albumin creatinine ration; eGFR, estimated glomerular filtration rate; CKD, chronic kidney disease; KDIGO, Kidney Disease: Improving Global Outcomes; n, total number of Ghanaians living in Amsterdam, Berlin and London among the various levels of PS constructs; PS, psychosocial stressors; %, proportion of individuals with CKD among the various levels of PS constructs in Europe.

assertions are supported by several studies. Specifically, these studies have shown that social support positively affect health outcomes through mechanisms such as increased patient compliance with therapies, decreased levels of depressive affect, direct physiological effects on the immune system and improved perception of quality of life.[58 59] The lack of association observed in this study may also be attributed to other mechanisms, which influence the associations between PS and CKD. Another reason for the lack of association between PS and CKD in this study could be the cross-sectional analysis of our study. The association between PS and CKD has been shown to be cumulative and builds over substantial period of time.[62] To effectively evaluate this, longitudinal study design is required. This suggests the need for more longitudinal studies in future research in assessing the associations between PS and CKD outcomes.[62 63]

### Strengths and limitations

Our study is the first to use all four robust constructs of PS to determine association between PS and CKD. This gave our study a more robust definition of PS compared with other similar studies. The use of all three definitions of CKD per KDIGO guidelines also provided a broader definition of CKD and allowed comparison between different geographical regions. The use of a homogenous population of Ghanaians and standardised protocols and diagnostic criteria in this study also provided a novel opportunity to compare Ghanaians living in rural and urban Ghana and their compatriots living in Europe. There are limitations to our study. The effect of PS on CKD has been reported to be cumulative and takes a long period of time, therefore the use of cross-sectional design prevented us from determining the longitudinal and cumulative effect of repeated exposure to PS among the two populations. PS is captured and experienced in different magnitude across different populations. We were unable to ascertain if PS as defined in this study was adequately captured among Ghanaians living in rural and urban Ghana. Lastly, the four PS measures were assessed separately because of multicollinearity among the measures. There are several methods of addressing multicollinearity among measures such as partial least squares regression, principal component analysis, data reduction technique, which when used could have influenced the interpretation of the study results.

### CONCLUSION

We identified positive association between stress at work/home and albuminuria and CKD risk among Ghanaians living in Berlin. Conversely, our study shows no associations between stress as indicated by four PS indicators and prevalence of CKD. Consequently, there is the need to explore other factors that may be responsible for the observed differences in the prevalence of CKD among Ghanaians living in rural and urban Ghana and their peers living in Europe.

**Author affiliations**

[1]Department of Public Health, Academic Medical Center, University of Amsterdam, Amsterdam Public Health Research Institute, Amsterdam, The Netherlands

[2]Department of Medical Laboratory Sciences, School of Biomedical and Allied Health Sciences, College of Health Sciences, University of Ghana, Accra, Ghana

[3]Department of Medicine, School of Medicine and Dentistry, University of Ghana and Korle-Bu Teaching Hospital, Accra, Ghana

[4]Department of Non-Communicable Disease Epidemiology, London School of Hygiene and Tropical Medicine, London, United Kingdom

[5]Kumasi Centre for Collaborative Research, KNUST, Kumasi, Ghana

[6]Julius Global Health, Julius Center for Health Sciences and Primary Care, University Medical Centre, Utrecht University, Utrecht, The Netherlands

[7]Division of Epidemiology and Biostatistics, School of Public Health, Faculty of Health Sciences, University of the Witwatersrand, Johannesburg, South Africa

[8]Institute of Tropical Medicine and International Health, Charité–University Medicine Berlin, Berlin, Germany

[9]Department of Molecular Epidemiology, German Institute of Human Nutrition Potsdam-Rehbrücke, Nuthetal, Germany

[10]Institute for Social Medicine, Epidemiology and Health Economics, Charité–Universitaetsmedizin Berlin, Berlin, Germany

[11]Department of Endocrinology and Metabolism, Charité–University Medicine Berlin, Berlin, Germany

[12]German Centre for Cardiovascular Research (DZHK), Berlin, Germany

[13]Center for Cardiovascular Research (CCR), Charité–University, Medicine, Berlin, Germany

[14]MKPGMS, Uganda Martyrs University, Kampala, Uganda

**Acknowledgements** The authors would like to thank the research assistants, interviewers and other staff of the five research locations who took part in gathering the data and the Ghanaian volunteers in all the participating RODAM sites. The authors would like to thank the advisory board members for their valuable support in shaping the RODAM study methods and the Academic Medical Centre Biobank for their support in biobank management and high-quality storage of collected samples.

**Contributors** The coauthors have all contributed substantially to this manuscript and approve of this submission. Research idea and study design: DNA, CA, KS, DA, EB, KM, JA; data acquisition and curation: DNA, CA, EB, KM; data analysis/interpretation: DNA, CA, KS, DA, EB, KM, LS, JA, EO-D, KK-G, FM, MS, ID, JS, SKB; statistical analysis: DNA, CA, KS. DNA, CA, KS, DA, EB, KM, LS, JA, EO-D, KK-G, FM, MS, ID, JS, SKB; contributed important intellectual content during manuscript drafting or revision and accepts accountability for the overall work by ensuring that questions pertaining to the accuracy or integrity of any portion of the work are appropriately investigated and resolved. DNA and CA takes responsibility that this study has been reported honestly, accurately and transparently; that no important aspects of the study have been omitted and that any discrepancies from the study as planned have been explained.

**Funding** This work was supported by the European Commission under the Framework Programme (grant number: 278901). The Wellcome Trust supported Professor Smeeth's contribution (grant number WT082178). Professor Spranger was supported by the DZHK (German Center for Cardiovascular Research) and the Berlin Institute of Health (BIH).

**Competing interests** None declared.

**Patient consent for publication** Not required.

**Ethics approval** The ethics committees in Ghana, the Netherlands, Germany and the UK approved the study protocol.

**Provenance and peer review** Not commissioned; externally peer reviewed.

**Data sharing statement** Data are available from the RODAM research cohort, a third party. EB affiliated with the RODAM research cohort and a coauthor of this paper in accordance with the RODAM requirements for collaboration. EB is the Data Collection Coordinator of RODAM and may be contacted with further questions (e. j.beune@amc.uva.nl). Additionally, researchers interested in further collaboration with RODAM may see the following URL: http://www.rod-am.eu/

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
