## [Reviewer comments · BMJ Open]

ARTICLE DETAILS

TITLE (PROVISIONAL)	A CROSS-SECTIONAL STUDY OF ASSOCIATION BETWEEN PSYCHOSOCIAL STRESSORS WITH CHRONIC KIDNEY DISEASE AMONG MIGRANT AND NON-MIGRANT GHANAIS LIVING IN EUROPE AND GHANA: THE RODAM STUDY
AUTHORS	Adjei Nana, David; Stronks, Karien; Adu, Dwomoa; Beune, Erik; Meeks, Karlijn; Smeeth, Liam; Addo, Juliet; Owusu-Dabo, Ellis; Klipstein-Grobusch, Kerstin; Mockenhaupt, Frank; Schulze, Matthias; Danquah, Ina; Spranger, Joachim; Bahendeka, Silver; Agyemang, Charles

VERSION 1 – REVIEW

REVIEWER	Joseph Lunyera Duke University School of Medicine
REVIEW RETURNED	12-Dec-2018

GENERAL COMMENTS	This is a novel study that examines a very interesting hypothesis - it juxtaposes health outcomes of Ghanaian migrants in Europe to that of their peers in Ghana. I applaud the authors for this brilliant approach. That said, I feel the study needs several critical improvements before the data can contribute meaningful insights on this important topic: 1) Statistical analysis: Since the psychosocial measures were highly correlated with each other in this study, the analysis should account for this multicollinearity. 2) Results: The reporting of the results does not comply with STROBE guidelines. Specifically, the authors should include numbers in the text (which would facilitate readership - one does not have to always refer to the Table). 3) Discussion: a) The authors posit that psychosocial stressors lead to adverse kidney outcomes via CKD risk factors such as hypertension and diabetes. While this is partly true, the discussion (and may be the introduction as well) needs to consider other direct mechanisms for this link. These pathways are extensively discussed in the literature, including some of the references cited in this manuscript (e.g., see reference #5: Bruce MA, et al. J Investig Med. 2009).
--

	b) a critical omission from the discussion that needs consideration is the fact that stress exposures have cumulative adverse influences that impact health over a long period of time, and as such, can only be evaluated using longitudinal study designs. Granted, the authors acknowledged the limitations of their cross-sectional design. However, since this limitation is central to how we interpret the present data in the context of the relationship between stress exposures and adverse health, there needs to be a dedicated paragraph discussing this.
--	---

REVIEWER	Loretta Cain Assistant Professor John D. Bower School of Population Health University of Mississippi Medical Center Jackson, MS USA
REVIEW RETURNED	25-Dec-2018

GENERAL COMMENTS	The overall concept of the paper is good. However, see the following concerns.  1. The writing needs significant edits, especially in the abstract. 2. There are too many tables that add little or nothing to the overall results. 3. There are stratified tables with no tests for interactions. 4. There are p-values for Table without reference as to how the p-values were obtained. 5. This seems to be a secondary data analysis but patient recruitment is also mentioned. The reviewer provided a marked copy with additional comments. Please contact the publisher for full details.
---

VERSION 1 – AUTHOR RESPONSE

Reviewer: 1

Reviewer Name: Joseph Lunyera

Comment 1: Since the psychosocial measures were highly correlated with each other in this study, the analysis should account for this multicollinearity.

Response 1: Thank you for this important comment. Because psychosocial measures were highly correlated, the psychosocial measures were assessed separately and thus preventing multicollinearity.

Comment 2: The reporting of the results does not comply with STROBE guidelines. Specifically, the authors should include numbers in the text (which would facilitate readership - one does not have to always refer to the Table).

Response 2: Thank you for this important comment. We have modified the reporting of all statistics contained in the manuscript in accordance with STROBE guidelines Pg. 11-23.

Comment 3: The authors posit that psychosocial stressors lead to adverse kidney outcomes via CKD risk factors such as hypertension and diabetes. While this is partly true, the discussion (and may be the introduction as well) needs to consider other direct mechanisms for this link. These pathways are

extensively discussed in the literature, including some of the references cited in this manuscript (e.g., see reference #5: Bruce MA, et al. J Investig Med. 2009).

Response 3: Thank you for this important comment. We have modified the introduction Pg. 5, lines 206-214; and discussion section Pg. 25; lines 652-658 to reflect other direct mechanisms of the link between psychosocial stressors and kidney outcomes.

Comment 4: A critical omission from the discussion that needs consideration is the fact that stress exposures have cumulative adverse influences that impact health over a long period, and as such, can only be evaluated using longitudinal study designs. Granted, the authors acknowledged the limitations of their cross-sectional design. However, since this limitation is central to how we interpret the present data in the context of the relationship between stress exposures and adverse health, there needs to be a dedicated paragraph discussing this.

Response 4: We have discussed the cumulative effect of stress on health over substantial period and have presented its limitation in this study. Pg. 25; lines 652-658; Pg. 26, lines 669-670.

Reviewer: 2

Reviewer Name: Loretta Cain

Institution and Country: Assistant Professor

Comment 1: The writing needs significant edits, especially in the abstract.

Response 1: We have edited the abstract and all the other sections of the manuscript to conform with that of the journals guideline. Pg. 3, lines 90-116..

Comment 2: There are too many tables that add little or nothing to the overall results.

Response 2: We stratified and added these tables to account for the presence of interaction among the various sites of the study. Also, to answer our second objective.

Comment 3: There are stratified tables with no tests for interactions.

Response 3: We have provided tests for interactions responsible for the stratification of our tables.

Comment 4: There are p-values for Table without reference as to how the p-values were obtained.

Response 4: Reference to how p-values were obtained have been provided. Pg. 10 lines 382-384 and Pg/line 11, lines 402-404.

Comment 5: This seems to be a secondary data analysis but patient recruitment is also mentioned.

Response 5: This study collected primary data from all sites and patients were recruited into the study. No secondary data was used in this study.

VERSION 2 – REVIEW

REVIEWER	Loretta Cain University of Mississippi Medical Center, USA
REVIEW RETURNED	01-Apr-2019

GENERAL COMMENTS	I appreciate your work assessing the relationship between psychosocial risk factors and kidney disease outcomes. You did a good job framing the problem, stating the research question, and using appropriate methods in the analysis. I only have a few comments.  1. There were several variables that were not clearly defined: educational level, CKD risk, and smoking status. 2. You stated that you used a z-test to assess the relationship between categorical variables and site. Z-tests are used to assess means. Therefore, z-tests cannot be used to assess the association between 2 categorical variables. Please clarify/correct. 3. Last, please provide a rationale for stratifying disease risk factors. Were interaction tests completed before stratifying? The reviewer provided a marked copy with additional comments. Please contact the publisher for full details
--

REVIEWER	Joseph Lunyera Duke University School of Medicine
REVIEW RETURNED	06-Apr-2019

GENERAL COMMENTS	I acknowledge the revisions and feel that they have greatly improved the rigor of the study. I have two minor comments for consideration. First, the revised results reported in the abstract suggest that this study is not entirely a negative one as the authors seem to infer in their conclusions (stress at work/home was associated with albuminuria and CKD risk profiles; these are important positive findings that should be masked in the larger narrative of this being a negative study). I think the conclusion should be balanced to reflect this heterogeneity in findings. My second comment is that I disagree with the approach that the authors used to address the issue of multicollinearity among the stress measures. Running separate regression models for each member of a set of correlated variables is only valid if the correlation arises from similarities in data structure and not due to underlying mechanisms by which the correlated independent variables associate with the dependent variable. However, the literature consistently shows that psychosocial factors are correlated with each other because they capture intertwined dimensions of our daily lived experiences that work through certain common physiologic and behavioral pathways to affect health. Thus, a better way to deal with this correlation issue is to include all these variables in the same model while accounting for the correlation among them using data reduction technique. This does not have to be implemented in the present study given that it is
---

	already in an advanced stage but at least should be discussed in the limitations to the interpretation of the data presented here.
--	--

VERSION 2 – AUTHOR RESPONSE

Reviewer: 2

Comment 1: There were several variables that were not clearly defined: educational level, CKD risk, and smoking status.

Response 1: Thank you for this comment. We have clearly defined all the variables in the manuscript. Pg. 7 lines 274-282.

Comment 2: You stated that you used a z-test to assess the relationship between categorical variables and site. Z-tests are used to assess means. Therefore, z-tests cannot be used to assess the association between 2 categorical variables. Please clarify/correct.

Response 2: We have clarified the use of z-test in the manuscript and amended that section (Z-test for proportion). There are z-tests for both continuous and categorical outcomes as stated in most statistical softwares (STATA) and several papers. Pg. 10 line 379

Comment 3: Last, please provide the rationale for stratifying disease risk factors. Were interaction tests completed before stratifying?

Response 3: We have provided rationale for stratifying the disease risk factors. Yes, interaction tests were completed before stratifying. Yes interaction tests were performed and this necessitated our stratification of disease risk factors. We have provided a rationale for this in the methods section of the manuscript. Pg. 11 line 397-398

Reviewer: 1

Comment 1: Please state any competing interests or state 'None declared': None declared

Response 1: We have provided a statement on competing interest. We do not have any to declare ("None"). Pg. 27 line 652

Comment 2: First, the revised results reported in the abstract suggest that this study is not entirely a negative one as the authors seem to infer in their conclusions (stress at work/home was associated with albuminuria and CKD risk profiles; these are important positive findings that should be masked in the larger narrative of this being a negative study). I think the conclusion should be balanced to reflect this heterogeneity in findings.

Response 2: Thank you for this important comment. We have modified the conclusion section of the abstract and the discussion sections to reflect the findings and have balanced the heterogeneity in our findings. Pg. 3 lines 114-116 and Pg. 26 lines 614-615

Comment 3: My second comment is that I disagree with the approach that the authors used to address the issue of multicollinearity among the stress measures. Running separate regression models for each member of a set of correlated variables is only valid if the correlation arises from similarities in data structure and not due to underlying mechanisms by which the correlated independent variables associated with the dependent variable. However, the literature consistently shows that psychosocial factors are correlated with each other because they capture intertwined

dimensions of our daily lived experiences that work through certain common physiologic and behavioral pathways to affect health. Thus, a better way to deal with this correlation issue is to include all these variables in the same model while accounting for the correlation among them using data reduction technique. This does not have to be implemented in the present study given that it is already in an advanced stage but at least should be discussed in the limitations to the interpretation of the data presented here.

Response 3: We thank the reviewer for providing an alternative method for dealing with multicollinearity among the stress measures. We agree with the reviewer on the fact that there are several methods of assessing multicollinearity among measures. We have therefore discussed this in the limitations section of the study. Pg. 26 lines 606-610.